# Revealing Virtual Water Transfers and Imbalanced Economic Benefits Hidden in China's Interprovincial Trade

**Jie Zheng** [1,2], **Sanmang Wu** [1,2,*], **Li Li** [1,2], **Shantong Li** [3], **Qiuping Li** [1,2] **and Qi An** [1,2]

1   School of Economics and Management, China University of Geosciences, Beijing 100083, China;
    18810595057@163.com (J.Z.); lili@cugb.edu.cn (L.L.); liqiuping2017@outlook.com (Q.L.);
    aq199919@126.com (Q.A.)
2   Key Laboratory of Carrying Capacity Assessment for Resource and Environment,
    Ministry of Natural Resources, Beijing 100083, China
3   Development Research Center of State Council, Beijing 100010, China; shantongdrc@163.com
*   Correspondence: wusanmang@sina.com

**Abstract:** Interprovincial trade has expanded China's virtual water consumption and economic development. This study uses an environmental–economic inequality index to calculate the virtual water and economic benefit transfer imbalances in interprovincial trade and applies a structural path analysis (SPA) model to find the imbalances on the key virtual water supply chain paths between provinces. The findings are fourfold. (1) The developed provinces, such as Guangdong, Jiangsu, and Shandong, had more virtual water on the consumption side from 2002 to 2017 and had the most value added on the consumption and production sides. (2) The developing provinces in northwest and central China suffered from net virtual water outflows and negative value-added gains in bilateral trade with developed provinces. (3) The developed provinces, such as Beijing, acquired more virtual water from other provinces in 0–5 production tiers, but only a small part of the value added was transferred out. (4) All of the four top ranking virtual water supply chain paths of Beijing came from other provinces, accounting for 28.22% of the total virtual water flowing to Beijing, but their value added only accounted for 1.44%. It is suggested that provinces adopt differentiated water-use systems to reduce virtual water transfer imbalances and provide subsidies to the nodes to compensate the economic benefits on key virtual water supply chain paths.

**Keywords:** multiregion input–output model; structural path analysis; China's interprovincial trade; virtual water transfers; value added

## 1. Introduction

China is not only a nation facing water resource shortages in the world, but also a country with a considerable imbalance in regional water resources distribution. In 2017, China's water resources per capita were calculated as 2015 m$^3$/person [1], less than half of the global rate of water resources per capita (5725 m$^3$/person). In 2020, China's water resources per capita were calculated as 2239.8 m$^3$/person [2], whereas in the northern provinces, it was less than 1821.5 m$^3$/person (Figure 1). To alleviate shortages and the imbalanced distribution of water resources in China, the government has initiated multiple physical water transfer projects, such as the South-to-North Water Transfer Project, and it has implemented three stringent controlling redlines concerning water use [3]. Although physical water transfer projects have a positive influence in alleviating water resource shortages in some regions, they also have a potentially negative impact on the ecosystem [4].

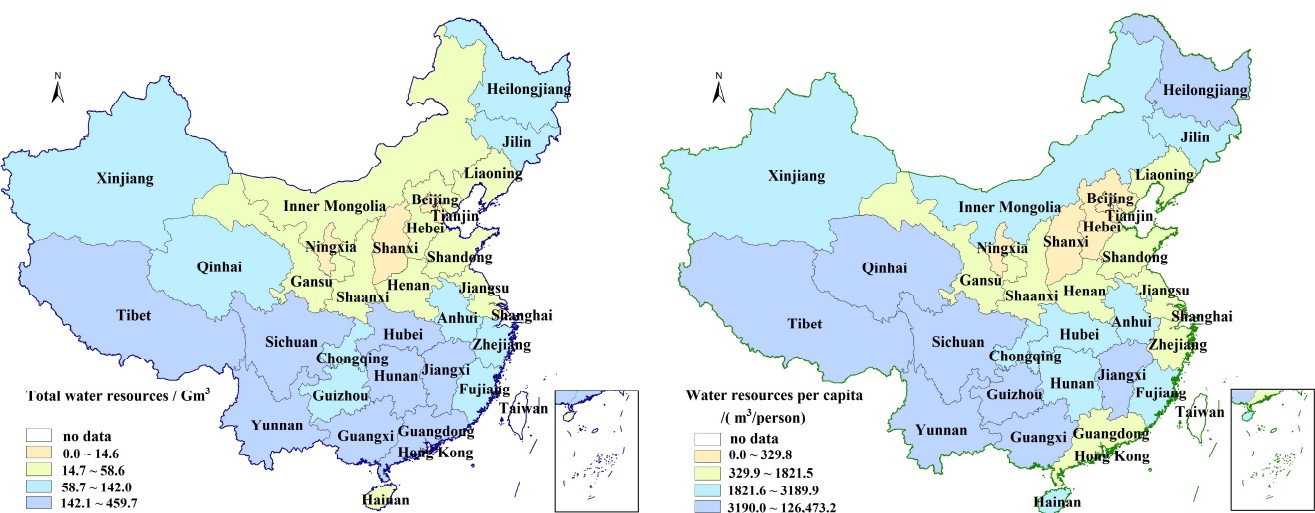

**Figure 1.** Total water resources and water resources per capita of Chinese provinces in 2020.

Trade is an important driving force in promoting provinces' economic development and is conducive to leveraging comparative advantages and advancing the development of the professional division of labor in provinces [5,6]. Trade is not only accompanied by the exchange of commodities and currency, but also the exchange of pollutants, resources, and money [7–9]. Studies have demonstrated that large amounts of virtual water are transferred with the development of trade [10–12]. The virtual water transfers associated with trade can redistribute water resources and represent an important means for alleviating water resource shortages in water-deficient provinces [13–16]. However, while trade promotes provinces' development, the dilemma regarding the imbalanced distribution of virtual water and economic benefits remains [17–19].

Interprovincial transfers of virtual water and economic benefits are key concerns, and researchers have attempted to identify the key provinces, sectors, and processes [19,20]. Input–output (IO) models divide virtual water and value-added transfers into direct and indirect transfers [21]. According to the number of regions analyzed, IO models are divided into single-region input–output and multiregion input–output (MRIO) models. MRIO models are often used to track local, external, and demand-driven virtual water and value-added transfers [19,22–26]. For example, Zhang et al. [25] used an MRIO model, finding that water-deficient provinces in north China benefited from the net inflows of virtual water in 2012 by outsourcing water-intensive products from other provinces, whereas water-deficient provinces in the northwest suffered from net outflows of virtual water. Xin et al. [19] used an MRIO model, revealing that up to 70% of demand-driven virtual water in developed provinces (Beijing, Tianjin, and Shanghai) came from other provinces in 2015, whereas the transferred added value accounted for only about 40%. Moreover, when some developing provinces, such as Xinjiang and Heilongjiang, traded with developed provinces, there were not only net outflows of virtual water, but also net outflows of value added; however, from the initial producers to the final consumers, MRIO models are unable to allocate indirect virtual water and value added to intermediate consumers in different production tiers [27]. Moreover, with the deepening of modern industrial divisions of labor, virtual water transfer paths and value added in different production tiers may be very complex, but they are necessary to track. Accordingly, by decomposing a total multiplier effect into the effects of each production tier and ranking the results, structural path analysis (SPA) methods can be applied to analyze the key nodes and supply chain paths that drive environmental burdens and resource use [20,28,29].

With the ability to track the environmental burdens and resource transfers hidden in regional and interprovincial trade, researchers are already aware of the environmental dilemmas and imbalanced economic benefits in regions and provinces. Some studies have investigated the economic and environmental inequalities hidden in trade, such

as air pollutants [30,31], carbon emissions [32,33], water pollutants [18], and virtual water [19,34]. These studies consistently demonstrated that when engaging in bilateral trade with developed regions or provinces, developing regions or provinces not only experienced net pollutant inflows, but also net value-added outflows, thus bearing environmental and economic inequality. To measure inequality between air pollutants and value-added transfers hidden in China's interprovincial trade, Zhang et al. [30] developed a regional environmental inequality (REI) index. The REI index can integrate specific pollutants and value-added trade flows, and it clearly illustrates the circumstances of each province in bilateral trade. Chen et al. [34] examined the virtual water and land inequality embodied in international trade by applying the REI index. Similarly, Xiong et al. [18] applied the REI index to measure the water pollutant inequality embodied in China's interprovincial trade, using the SPA method to track the top 50 key water pollutants' flow paths across sectors. They found that main paths were generated by processing agricultural products in inland provinces into food for the consumption of developed provinces in the southeast. Xiong et al. [18] measured only virtual water on main supply chain paths but did not measure the economic benefits on the paths that reveal inequality. In the existing studies, researchers have examined the imbalance of virtual water and economic benefit transfers in interprovincial bilateral trade, but imbalances hidden in virtual water supply chain paths are rarely studied.

This study calculated the virtual water transfers and value added of 30 provinces in China from 2002 to 2017, analyzed net transfers of virtual water and value added of provinces in 2017, and used an environmental–economic inequality index to calculate the degree of imbalance in interprovincial bilateral trade. Moreover, this study calculated 30 provinces' virtual water and value added that went through 0–5 production tier transfers in 2017 and analyzed virtual water transfers and economic benefits disparities on main supply chain paths of virtual water in two case provinces. An in-depth study of virtual water transfers and economic benefits imbalances hidden in China's interprovincial trade is of considerable significance to scientifically formulate China's water resources management policies and strategically promote coordinated interprovincial development.

## 2. Methodology

### 2.1. Multiregion Input–Output Analysis

The multiregion input–output (MRIO) model is an economic–mathematical approach used to analyze the flows of money or resources between different provinces and sectors driven by demand [35]. In Equation (1), it is assumed that there are $m$ provinces and $n$ sectors, in which $r$ and $s$ represent provinces' inflow and outflow, respectively, and $i$ and $j$ represent sectors' inflow and outflow, respectively. Here $x$ and $y$ are mn $\times$ 1 vectors, with their respective elements, $x_i^r$, representing the output of sector $i$ in province $r$ and $y_i^{rs}$ representing the products of sector $i$ from province $r$ consumed finally in province $s$. $A = \left(a_{ij}^{rs}\right)_{mn \times mn}$ is the intermediate input coefficient matrix with its element, $a_{ij}^{rs} = \frac{z_{ij}^{rs}}{x_i^r}$, where $z_{ij}^{rs}$ indicates intermediate input from sector $i$ in province $r$ to sector $j$ in province $s$. $I$ is an mn $\times$ mn identity matrix, and $(I - A)^{-1}$ is the Leontief inverse matrix.

$$x = (I - A)^{-1} \times y \tag{1}$$

In Equations (2) and (3), virtual water flows between provinces and sectors can be calculated. Here, $f = \left(f_i^r\right)_{mn \times 1}$ and $d = \left(d_i^r\right)_{mn \times 1}$ denote water-use intensity and value-added coefficient vectors with their respective elements, $f_i^r = \frac{w_i^r}{x_i^r}$ and $d_i^r = \frac{v_i^r}{x_i^r}$, where $w_i^r$ and $v_i^r$ represent the water use and value added of sector $i$ in province $r$. Where $f^r$ and $f^s$ represent water-use intensity coefficient vectors of province $r$ and province $s$, but zeroes for all other provinces ($r \neq s$), $y^s$ and $y^r$ represent the final demand vectors of province $s$ and province $r$, with zeroes for all other provinces ($r \neq s$). The symbol ^ indicates the diagonalization matrix of the corresponding vectors. $W^{rs}$ represents the virtual water

flowing from province $r$ to province $s$, and $W^{sr}$ represents the virtual water flowing from province $s$ to province $r$.

$$W^{rs} = \hat{f}^r (I - A)^{-1} \hat{y}^s \tag{2}$$

$$W^{sr} = \hat{f}^s (I - A)^{-1} \hat{y}^r \tag{3}$$

$$W^{rs}_{net} = W^{rs} - W^{sr} \tag{4}$$

where the matrix $W^{rs}_{net}$ refers to the net virtual water flows from province $r$ to province $s$. Note that there are both positive and negative values in $W^{rs}_{net}$. The positive value indicates net transfers from province $r$ to province $s$, and the negative value indicates net transfers from province $s$ to province $r$. $\overline{W^{rs}_{net}}$ is defined as the matrix with all positive values of net virtual water flows among the provinces for the following calculation of the WVI index:

$$W^r_{production} = \sum_{s=1}^{m} \hat{f}^r (I - A)^{-1} \hat{y}^s \tag{5}$$

$$W^r_{consumption} = \sum_{s=1}^{m} \hat{f}^s (I - A)^{-1} \hat{y}^r \tag{6}$$

In Equations (5) and (6), the matrices $W^r_{production}$ and $W^r_{consumption}$ represent the virtual water on the production and consumption sides in province $r$, respectively.

Similarly, the value-added flows between provinces and sectors can be calculated in Equations (7) and (8). $d^r$ and $d^s$ represent the value-added coefficient vectors of province $r$ and province $s$, but zeroes for all other provinces ($r \neq s$).

$$V^{rs} = \hat{d}^r (I - A)^{-1} \hat{y}^s \tag{7}$$

$$V^{sr} = \hat{d}^s (I - A)^{-1} \hat{y}^r \tag{8}$$

$$V^{rs}_{net} = V^{rs} - V^{sr} \tag{9}$$

where matrix $V^{rs}$ represents the added value flowing from province $s$ to province $r$, the matrix $V^{sr}$ represents the added value flowing from province $r$ to province $s$, and the matrix $V^{rs}_{net}$ refers to the net value-added flows from province $s$ to province $r$.

$$V^r_{production} = \sum_{s=1}^{m} \hat{d}^r (I - A)^{-1} \hat{y}^s \tag{10}$$

$$V^r_{consumption} = \sum_{s=1}^{m} \hat{d}^s (I - A)^{-1} \hat{y}^r \tag{11}$$

In Equations (10) and (11), the matrices $V^r_{production}$ and $V^r_{consumption}$ represent the value added on the production and consumption sides in province $r$.

*2.2. Interprovincial Virtual Water and Value-Added Transfer Imbalances Index*

This study applies a virtual water and value-added transfer imbalances index (WVI) to evaluate the imbalances associated with interprovincial trade. We assume there is $\forall B_{m \times m}$ with its element $b$, all of which are normalized to a range between 0 and 1 using following general equation:

$$f(b) = \frac{b - b_{min}}{b_{max} - b_{min}} \tag{12}$$

where $b_{min}$ and $b_{max}$ represent the minimum maximum and values of $b$, respectively.

After normalization, the WVI index matrix's element $wvi^{rs}$ can be calculated as follows:

$$wvi^{rs} = \begin{cases} f\left(\frac{\overline{w}^{rs}}{\overline{v}^{rs}}\right), \; if \; \overline{w}^{rs} > 0 \; and \; \overline{v}^{rs} > 0 \\ f(\overline{w}^{rs}) + f(\,|\overline{v}^{rs}|) + 1 \,, \; if \; \overline{w}^{rs} > 0 \; and \; \overline{v}^{rs} < 0 \end{cases} \tag{13}$$

Because $\overline{w}^{rs} > 0$, as stated in the definition of $\overline{W^{rs}_{net}}$, there are two types of relationships between province $r$ and province $s$. When $\overline{w}^{rs} > 0$ and $\overline{v}^{rs} > 0$, the virtual water transfer occurs from province $r$ to province $s$, and the value-added transfer occurs from province $s$

to province $r$. When $\overline{w}^{rs} > 0$ and $\overline{v}^{rs} < 0$, both virtual water and value-added transfers occur from province $r$ to province $s$. $\overline{w}^{rs}$ and $|\overline{v}^{rs}|$ are normalized, then added up to reflect imbalances. Moreover, these values are added up 1 to distinguish the results from the above. We note that the WVI value represents the relative degree of imbalance between a pair of provinces, wherein higher WVI values indicate a higher level of imbalance in transfers of virtual water and value added embodied in trade.

### 2.3. Structural Path Analysis

The power series expansion of the Leontief inverse matrix can split a virtual water supply chain into infinite paths, allowing us to identify the key nodes and paths by sorting the virtual water of each path.

$$(I - A)^{-1} = I + A + A^2 + A^3 + \ldots + A^n + \ldots \tag{14}$$

Therefore, the virtual water hidden in trade can be expanded into infinite tiers as follows:

$$W = \hat{f}(I-A)^{-1}\hat{y} = \underbrace{\hat{f}I\hat{y}}_{\text{Tier 0}} + \underbrace{\hat{f}A\hat{y}}_{\text{Tier 1}} + \underbrace{\hat{f}A^2\hat{y}}_{\text{Tier 2}} + \underbrace{\hat{f}A^3\hat{y}}_{\text{Tier 3}} + \cdots + \underbrace{\hat{f}A^n\hat{y}}_{\text{Tier n}} + \cdots \tag{15}$$

Similarly, the value added in trade can also be expanded as follows:

$$V = \hat{d}(I-A)^{-1}\hat{y} = \underbrace{\hat{d}I\hat{y}}_{\text{Tier 0}} + \underbrace{\hat{d}A\hat{y}}_{\text{Tier 1}} + \underbrace{\hat{d}A^2\hat{y}}_{\text{Tier 2}} + \underbrace{\hat{d}A^3\hat{y}}_{\text{Tier 3}} + \cdots + \underbrace{\hat{d}A^n\hat{y}}_{\text{Tier n}} + \cdots \tag{16}$$

By unraveling the $A$ matrix, the virtual water and value added of different production tiers driven by the demand of province $s$ can be decomposed into infinite paths:

$$
\begin{aligned}
W^s_{Tier0} &= \sum_{r=1}^m \sum_{i=1}^n f_i^r y_i^{rs} \\
W^s_{Tier1} &= \sum_{r,t=1}^m \sum_{i,j=1}^n f_i^r a_{ij}^{rt} y_j^{ts} \\
W^s_{Tier2} &= \sum_{r,t,u=1}^m \sum_{i,j,k=1}^n f_i^r a_{ij}^{rt} a_{jk}^{tu} y_k^{us} \\
W^s_{Tier3} &= \sum_{r,t,u,v=1}^m \sum_{i,j,k,l=1}^n f_i^r a_{ij}^{rt} a_{jk}^{tu} a_{kl}^{uv} y_l^{vs} \\
&\cdots
\end{aligned}
\tag{17}
$$

$$
\begin{aligned}
V^s_{Tier1} &= \sum_{r=1}^m \sum_{i=1}^n d_i^r y_i^{rs} \\
V^s_{Tier1} &= \sum_{r,t=1}^m \sum_{i,j=1}^n d_i^r a_{ij}^{rt} y_j^{ts} \\
V^s_{Tier2} &= \sum_{r,t,u=1}^m \sum_{i,j,k=1}^n d_i^r a_{ij}^{rt} a_{jk}^{tu} y_k^{us} \\
V^s_{Tier3} &= \sum_{r,t,u,v=1}^m \sum_{i,j,k,l=1}^n d_i^r a_{ij}^{rt} a_{jk}^{tu} a_{kl}^{uv} y_l^{vs} \\
&\cdots
\end{aligned}
\tag{18}
$$

where $f_i^r y_i^{rs}$ and $d_i^r y_i^{rs}$ refer to the virtual water and value added hidden in the final product, and the path is sector $i$ in province $r \to$ the demand of province $s$; $f_i^r a_{ij}^{rt} y_j^{ts}$ and $d_i^r a_{ij}^{rt} y_j^{ts}$ refer to the virtual water and value added that have undergone one production tier transfer, and the path is sector $i$ in province $r \to$ sector $j$ in province $t \to$ the demand of province $r$; and $f_i^r a_{ij}^{rt} a_{jk}^{tu} y_k^{us}$ and $d_i^r a_{ij}^{rt} a_{jk}^{tu} y_k^{us}$ refer to the virtual water and value added that have undergone two production tier transfers, and the path is sector $i$ in province $r \to$ sector $j$ in province $t \to$ sector $k$ in province $u \to$ the demand of province $r$. The paths of the subsequent tiers are similar.

*2.4. Data Sources*

The multiregion input–output tables for 2002, 2007, 2012, and 2017, including 42 sectors for 30 provinces in China, are used in this study [36–39]. Since China's input–output table is published every five years, the Chinese multiregion input–output table for 2017 is the latest table available at present. Considering the consistency of sectors across years, the MRIO tables with 42 sectors are combined into the MRIO tables with 28 sectors (Appendix A, Table A1 and Appendix B, Table A2).

The agricultural water use in each province can be obtained directly from China Statistical Yearbook [40–43], whereas there is no direct access to water use in each province's secondary or tertiary industrial sectors. This study draws on Chen et al. [24] to calculate the data. Here, water use in Chinese secondary or tertiary industrial sectors is divided by each sector's total output and multiplied by the output of each sector in each province, and then the ratio of the actual to the calculated total secondary or tertiary industrial water use in each province is used to correct them. The actual total secondary and tertiary industrial water use in each province can be obtained from China Statistical Yearbook [40–43].

The total water use of each Chinese secondary industrial sector in 2002 can be obtained from Chinese Environment Yearbook 2003 [44]; for 2007 and 2012, the data can be obtained from Annual Statistic Report on Environment in China 2007 and 2012 [45,46]. Lacking the data for 2017, this study assumes that the water use per unit value added in each sector remains constant, and the water use per unit value added of each sector in 2015 [47] is multiplied by the added value of each sector in 2017, and then divided by the price index in 2016 and 2015, where the added value of each sector in 2015 uses the MRIO data in 2012, which is multiplied by the price index in 2012, 2013, and 2014.

The total water use of each Chinese tertiary industrial sector in 2002 is available [48], whereas for 2007, 2012, and 2017, there is no direct access to the data. The data for other years is calculated by the data in 2002 being divided by the total tertiary industrial water use in 2002, and then being multiplied by the total tertiary industrial water use in each other year. The total tertiary industrial water use in each province can be obtained by subtracting the primary industry, secondary industry, and households in each province from the total water use in each province [40–43]. The household water use in each province can be calculated by the population proportion between each province and nation being multiplied by the total Chinese household water use [49–51], where, due to the lack of the data for 2017, the total Chinese household water use from 2015 is used.

## 3. Results and Discussion

*3.1. Accounting of Virtual Water and Value Added on the Production and Consumption Side*

The developed provinces had more virtual water on the consumption side and the agricultural provinces had more virtual water on the production side. From 2002 to 2017, Guangdong, Jiangsu, Zhejiang, Shandong, and Fujian had more virtual water on the consumption side, accounting for 33% of China's virtual water (Figure 2). Jiangsu, Xinjiang, Guangdong, Hunan, and Heilongjiang had the most virtual water on the production side, accounting for 38% of China's virtual water, due to the production of a large number of agricultural products. Of the 30 provinces in China, 15 obtained virtual water from other provinces through interprovincial trade. The primarily water-deficient northern or developed coastal provinces required external water resources or produced goods with high value added and low water use. For example, the virtual water of Liaoning and Zhejiang on the consumption side is 1.2 and 1.5 times that of the production side, respectively. In contrast, most of the remaining 15 provinces with virtual water outflows were agricultural provinces located in the central and western regions.

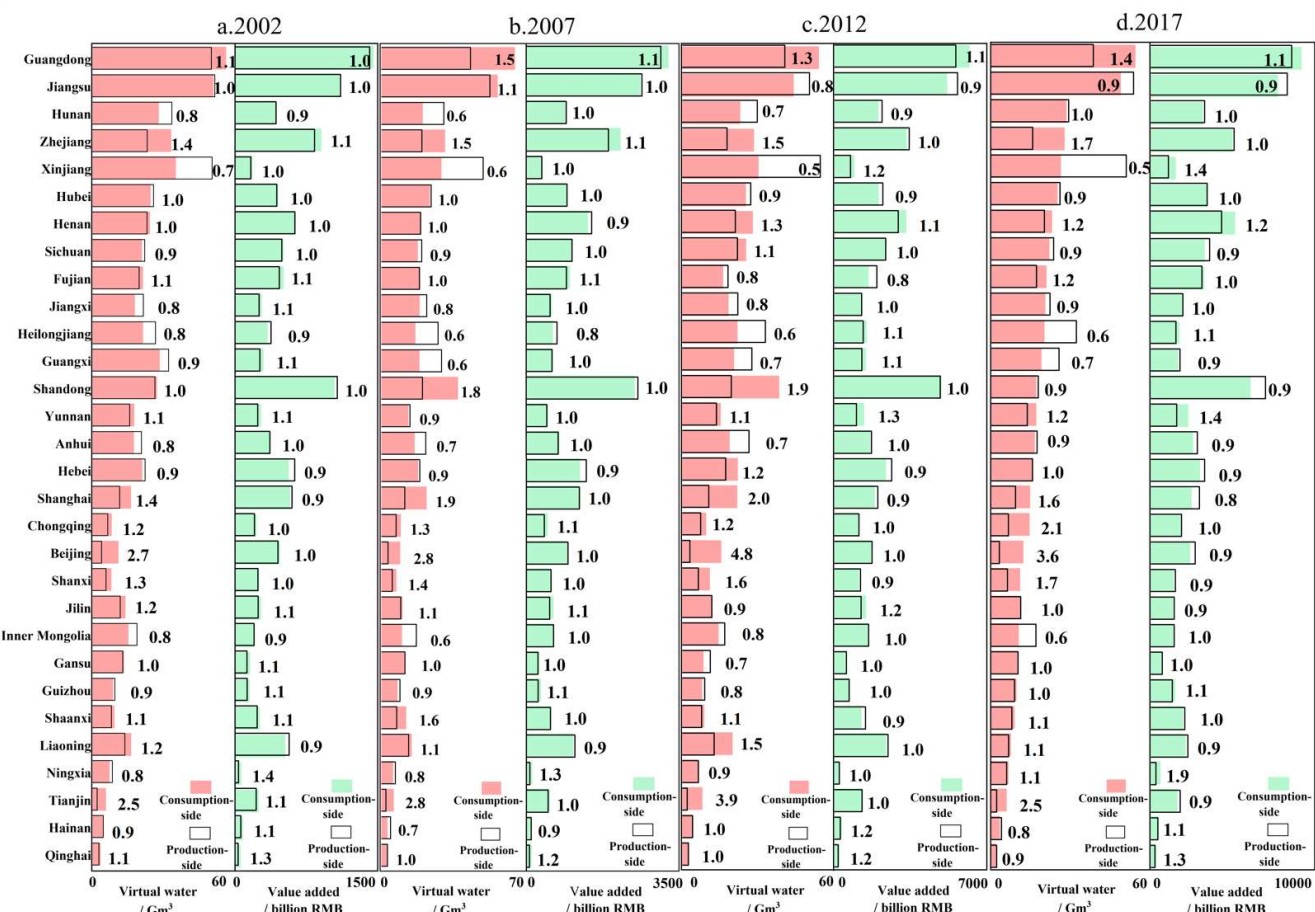

**Figure 2.** (**a**–**d**) show virtual water and value added of China's 30 provinces on the production and consumption sides in 2002, 2007, 2012, and 2017, respectively. Note: The numbers at the end of each bar represent the ratios of the consumption side to the production side. If the ratio of virtual water is greater than 1, the province has a net inflow of virtual water; if the ratio is less than 1, the province has a net outflow of virtual water. If the ratio of value added is greater than 1, the province has a net outflow of value added; if the ratio is less than 1, the province has a net inflow of value added.

Calculating the value-added component of trade, we found the developed provinces to have the most added value on the consumption and production sides. From 2002 to 2017, Guangdong, Jiangsu, Shandong, and Zhejiang had the most added value on the consumption and production sides, accounting for 36% and 35% of China's value added, respectively. Of the 30 provinces in China, 15 had net outflows of value added, such as Gansu and Yunnan. Their value added on the consumption side was 1.1 and 1.2 times that of the production side. Most of the provinces with net outflows of value added were developing provinces located in the northwestern or southwestern regions. A comparison of virtual water and value added revealed that the differences between the consumption and production sides of virtual water in the provinces were more striking than the value-added differences for most provinces. For instance, the range of consumption to production of virtual water ratios (0.5–4.8) was greater than the range of value-added ratios (0.8–1.9). The above mismatch may be due to the interprovincial trade of high-water-use and low value-added goods, such as agricultural products.

### 3.2. Net Transfers of Virtual Water and Value Added

Developed provinces were mainly win-win provinces, with net inflows of virtual water and value added, whereas most developing provinces were lose-lose provinces, with net outflows of virtual water and value added. The provinces in Group IV are predominantly developed provinces located in the north and on the east coast, which obtained net inflows of virtual water and net economic benefits (Figure 3). For instance, in 2017, Beijing and Shanghai, two of China's most developed provinces, had 9.42 and 5.6 Gm$^3$ net inflows of virtual water, representing 2.63 and 0.77 times the virtual water on their production side, respectively. At the same time, they had 302 and 472 billion RMB net inflows of value added, respectively. It could be that these provinces used a large number of high-water-use products from other provinces, while mainly high value-added products were produced locally. Most of the provinces in Group II are developing provinces in remote areas, with net outflows of virtual water and value added. For example, Xinjiang and Heilongjiang had 26.88 and 13.29 Gm$^3$ net outflows of virtual water and 462 and 212 billion RMB net outflows of value added, respectively. The provinces in Group III had net inflows of virtual water and net outflows of value added, including both developing and developed provinces, such as Henan, Yunnan, and Guangdong. Most provinces in Group I are agricultural provinces, some of which are relatively developed and close to those in Group IV, such as Jiangsu and Shandong. These provinces had net inflows of value added and net outflows of virtual water, likely due to resource endowments and geographical locations.

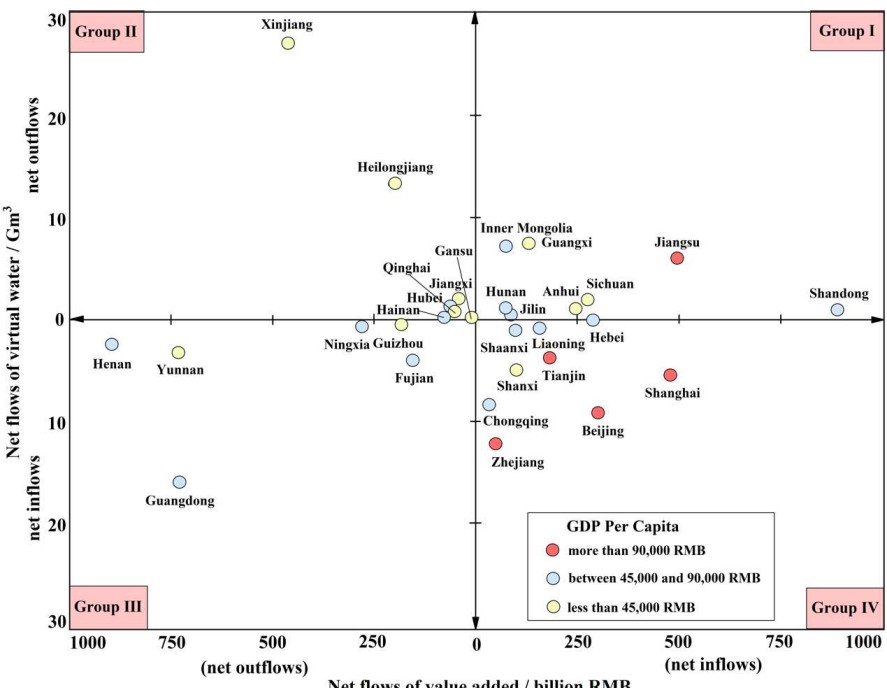

**Figure 3.** Classification of 30 provinces on the basis of net flows of virtual water and value added embodied in interprovincial trade in 2017.

Among the eight regions of China, net virtual water was primarily transferred from the northwest to the southwest, south coast, and Beijing–Tianjin (Figure 4a). In 2017, the net virtual water of the northwest accounted for about 70% of total virtual water outflows among the regions, which predominantly flowed out to the southwest and developed regions in China, including the south coast, Beijing–Tianjin, and the east coast. However, the net value added also transferred from the northwest and central regions to the north and east coast (Figure 4b). For example, as the main supplier of industrial products such as steel and electronics, about 43% of the net value added of the north and east coast came from the northwest and central regions.

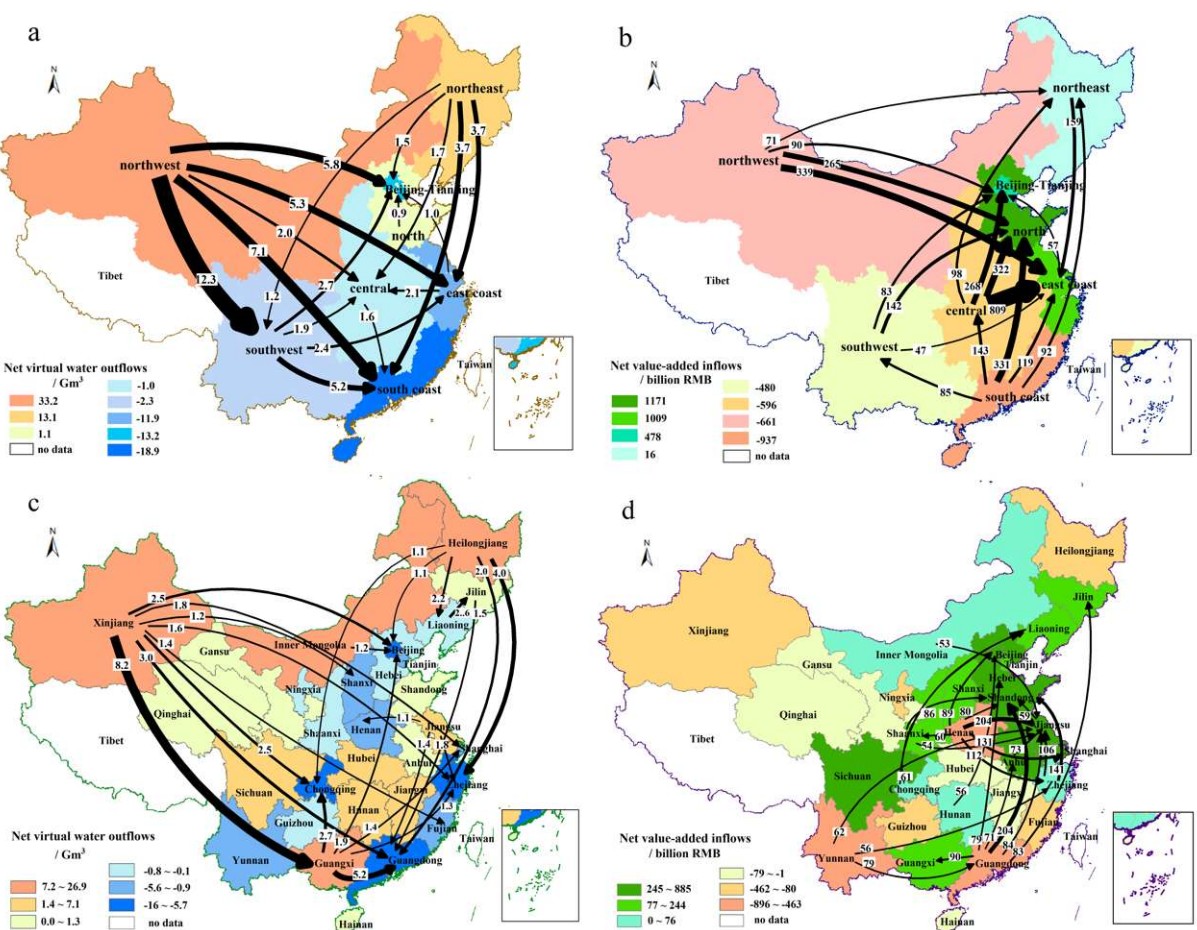

**Figure 4.** Net flows of virtual water and value added among the main provinces and regions. Note: The top 18 pairs of net flows of virtual water (**a**) with volumes larger than 0.9 Gm³ and value added (**c**) with sums larger than 47 billion RMB among the eight geographical regions are shown, accounting for 92.5% and 93.9% of the total outsourcing, respectively. The top 24 pairs of net flows of virtual water (**b**) with volumes larger than 1.1 Gm³ and value added (**d**) with sums larger than 53 billion RMB among the 30 provinces are presented, accounting for 51.9% and 30.3% of the total outsourcing, respectively.

In China's interprovincial trade, virtual water was primarily transferred from developing provinces to developed provinces (Figure 4c). For example, Xinjiang, Heilongjiang, and Guangxi had 26.9, 13.3, and 7.5 Gm³ net outflows of virtual water, respectively, accounting for about 68% of the total virtual water outflows among the provinces. Guangdong, Zhejiang, and Beijing are developed provinces located in the south coast, east coast and Beijing–Tianjin, respectively, with 16, 12.4, and 9.4 Gm³ net virtual water inflows, accounting for approximately 23%, 18%, and 13% of the total virtual water inflows among the provinces. Furthermore, the largest net value-added transfers occurred between the developed provinces (Figure 4d), such as Guangdong to Shandong (204 billion RMB), Jiangsu (494 billion RMB), and Shanghai (472 billion RMB). The developing provinces located in the northwest, central, and southwest regions were the main provinces with net value-added outflows. For example, Xinjiang, Henan, and Yunnan had 462, 896, and 729 billion RMB net outflows of value added, accounting for about 53% of the total interprovincial value-added outflows. Some coastal provinces (Shandong, Jiangsu) servicing cities (Beijing, Shanghai), industrial provinces (Hebei), and coal-rich provinces (Shanxi) obtained about 65% of their net value-added inflows via interprovincial trade.

### 3.3. Imbalances of Virtual Water and Economic Benefits in Bilateral Trade

Our results demonstrate that the developing provinces in China's northwest and central regions suffered from imbalances in net transfers of virtual water and economic benefits in bilateral trade with developed provinces, whereas the developed provinces benefited from the bilateral trade. For example, Xinjiang (WVI = 2.06) and Hubei (WVI = 1.62) both had 2.48 and 0.05 Gm$^3$ net outflows of virtual water in their bilateral trade with Beijing and 14.6 and 35.7 billion RMB net outflows of value added, respectively (Figure 5). WV > 1 indicates that net virtual water and value added were transferred from outflowing (primarily developing) provinces to the inflowing (primarily developed) provinces. There are 174 pairs of provinces with WVI > 1, accounting for approximately 40% of all the province pairs, such as Xinjiang–Zhejiang (WVI = 2.21), Xinjiang–Shanghai (WVI = 2.2), Inner Mongolia–Jiangsu (WVI = 1.95), and Henan–Tianjin (WVI = 1.68). The imbalance of net transfers borne by the developing provinces could be due to the current divisions of labor in China's value chain, as some developing and water-deficient provinces still choose to sacrifice scarce water resources in exchange for a small number of economic benefits. Additionally, the net virtual water of Jilin flowed out to Shaanxi, but its economic benefit inflow did not compensate for the net transfer of virtual water (WVI = 1). The WVI value of other pairs of provinces ranged from 0 to 0.01, indicating that economic benefits could compensate for net transfers of virtual water. We also revealed that imbalances between the long-distance provinces were more serious, which may be due to the transfers of virtual water in trade being affected by physical distance [52,53].

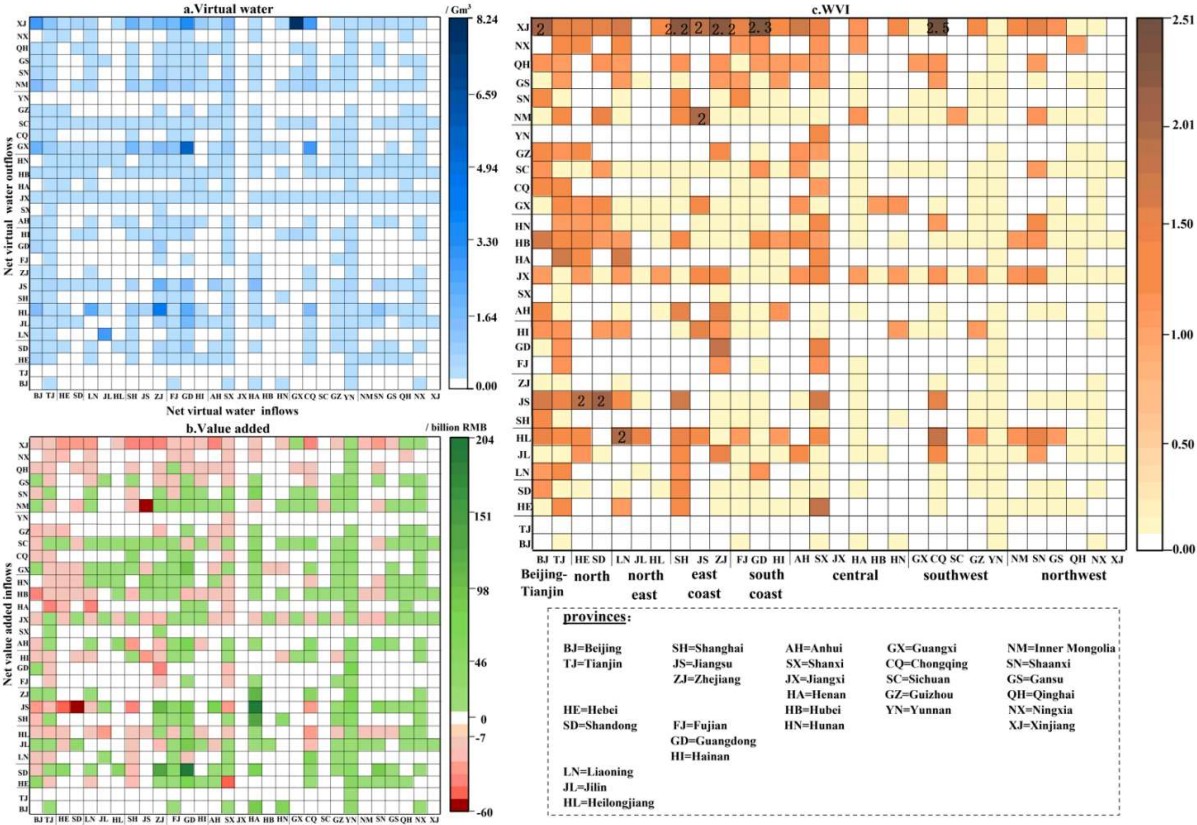

**Figure 5.** (**a**–**c**) show the transfer matrices of net virtual water, net value added, and WVI among the provinces in 2017, respectively. Note: When a pair of provinces has a WVI index value between 0 and 1, the province with a net outflow of virtual water has a net inflow of value added. When the value is between 1 and 2.51, the province bearing the imbalance has both a net outflow of virtual water and a net outflow of value added simultaneously. Darker colors signify higher levels of imbalance.

*3.4. Virtual Water Transfers and Imbalanced Economic Benefits in Different Tiers*

The virtual water in the T0, T1, T2, T3, T4, and T5 tiers accounted for 32%, 27%, 17%, 10%, 6%, and 4% of the total virtual water hidden in China's interprovincial trade in 2017, respectively. The value added in the T0, T1, T2, T3, T4, and T5 tiers accounted for 41%, 23%, 14%, 8%, 5%, and 3% of the total value added in China's interprovincial trade, respectively. T0 was the main tier of virtual water and value-added transfers, followed by T1 (Figure 6). The higher the production tier, the lower total virtual water and added value transferred in China's interprovincial trade. In China's interprovincial trade, the value added of the 30 provinces in T0 was the largest of all the tiers. Twenty-two provinces had the largest virtual water in T0, while the virtual water flowing to the T1 tier of Tianjin, Hebei, Jilin, Jiangsu, Zhejiang, Henan, Shaanxi, and Qinghai ranked as the largest of all tiers. In short, the main transfer paths of virtual water and value added embodied in China's interprovincial trade were relatively short.

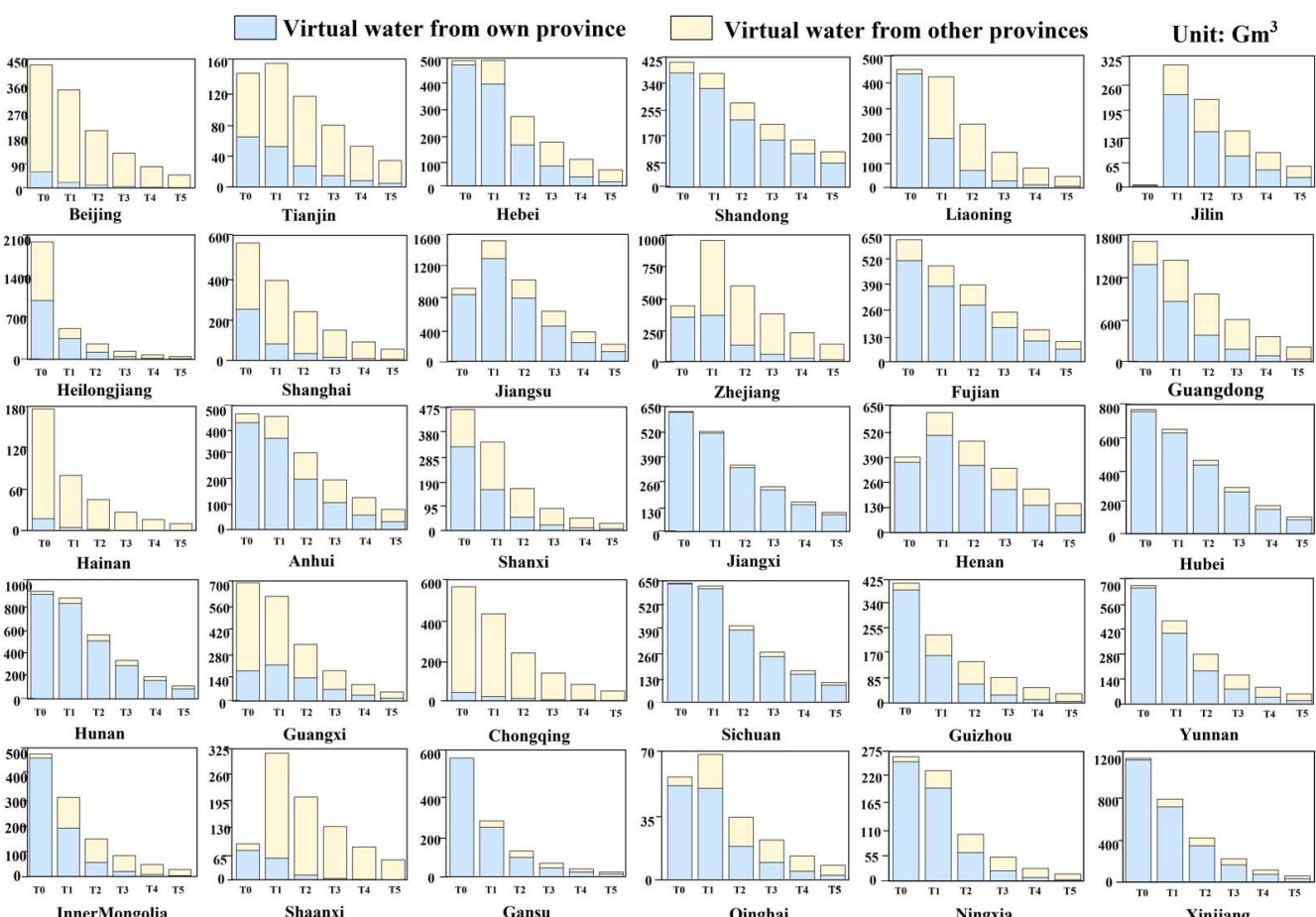

**Figure 6.** Virtual water in the first six tiers driven by the provinces' demand in 2017. Note: Virtual water in the T0 tier is directly transferred. Virtual water in the T1–T5 tiers has undergone 1–5 production tier transfers, respectively.

The developed provinces acquired more shares of virtual water supplied by other provinces in the first six tiers but transferred unequal shares of value added to other provinces (Figure 7), whereas developing provinces received less shares of virtual water from other provinces in the first six tiers, but transferred more shares of added value. For example, the virtual water to Beijing supplied by other provinces in the T0, T1, T2, T3, T4, and T5 tiers accounted for 86%, 94%, 95%, 96%, 97%, and 97% of the total virtual water of Beijing in each tier, respectively, which outweighed the proportion of value added (accounting for 19%, 41%, 54%, 64%, 72%, and 79%, respectively). The virtual water to

Guangdong supplied by other provinces in the first six tiers accounted for 19%, 41%, 61%, 71%, 78%, and 82%, respectively, outweighing the proportion of value added in each tier (accounting for 5%, 25%, 44%, 59%, 69%, and 77%, respectively). However, the virtual water to Xinjiang supplied by other provinces in the first six tiers accounted for 1%, 9%, 18%, 27%, 35%, and 44%, respectively, which is lower than the proportion of value added in each tier (26%, 48%, 62%, 72%, 79%, and 84%, respectively). The virtual water to Ningxia supplied by other provinces in the first six tiers accounted for 4%, 16%, 39%, 58%, 72%, and 82%, respectively, which is lower than the proportion of value added in each tier (33%, 66%, 82%, 90%, 95%, and 97%, respectively).

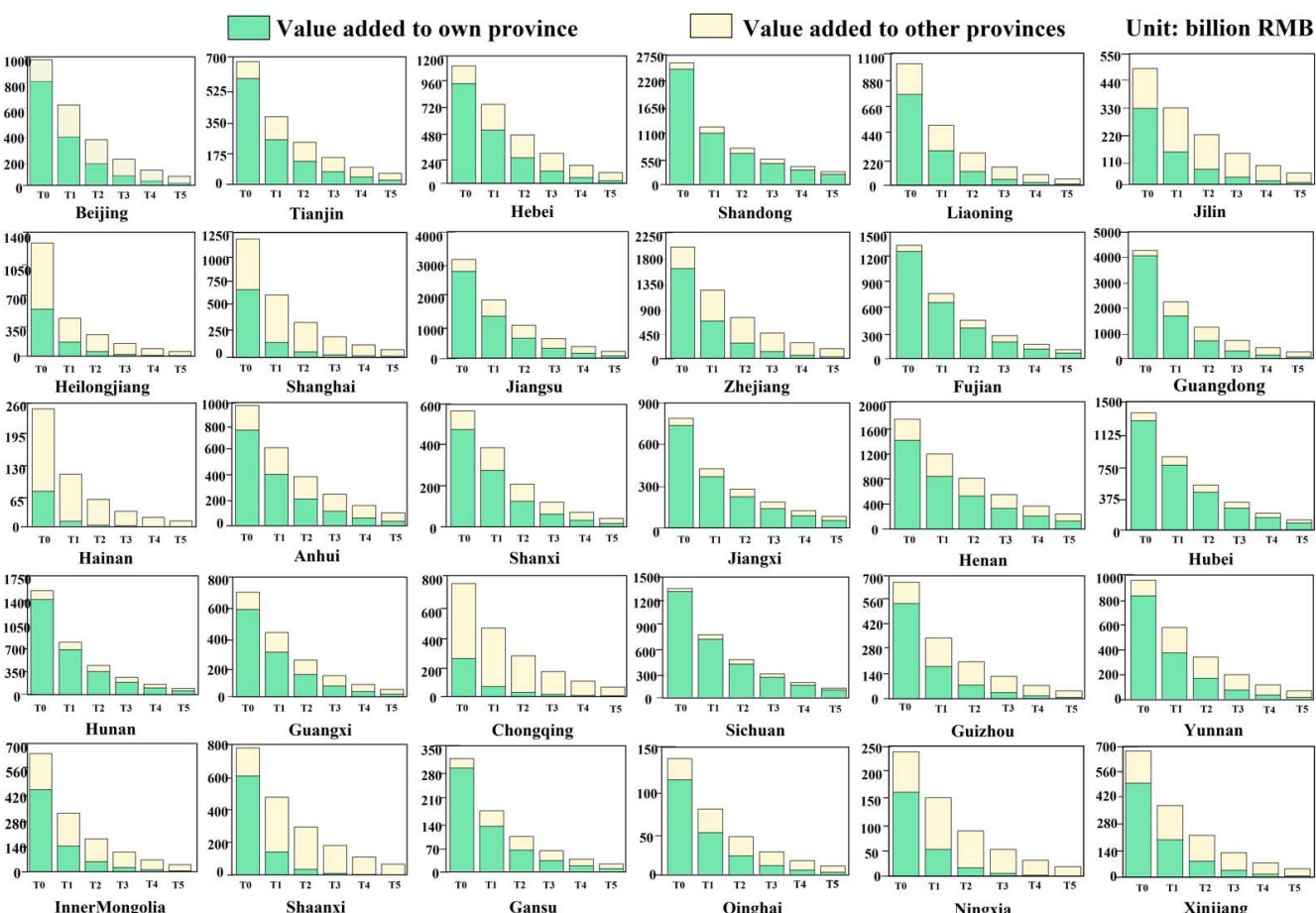

**Figure 7.** Value added in the first six tiers driven by the provinces' demand in 2017. Note: Value added in the T0 tier is directly transferred. Value added in the T1–T5 tiers has undergone 1–5 production tier transfers, respectively.

*3.5. Key Virtual Water Supply Chain Paths and Imbalanced Economic Benefits in Two Case Provinces*

It is well-known that Beijing and Ningxia are water-deficient provinces, and Beijing is a developed province located in Beijing–Tianjin while Ningxia is a developing province located in the northwest. The per capita water resources of Beijing and Ningxia in 2020 were 117.8 m$^3$/person and 153 m$^3$/person, respectively, less than one-tenth of China's water resources per capita (2239.8 m$^3$/person). The gross domestic product (GDP) per capita of Beijing was 164,889 RMB/person (National Bureau of Statistics of China, 2021), more than three times that of Ningxia (54,528 RMB/person). Of the 30 top ranking virtual water supply chain paths, virtual water flowing into Ningxia came from its own province (Figure 8). However, 23 paths of Beijing originated from 16 other provinces, accounting for about 92% of the total virtual water in the 30 top ranking paths, while the value added on the key virtual water supply chain paths only accounted for about 10%. Our results

demonstrated that the 30 top ranking key virtual water supply chain paths of Ningxia were all within its own province. The 30 top ranking key virtual water supply chain paths of Beijing were more diversified, while very few economic benefits were transferred to other provinces on the key paths.

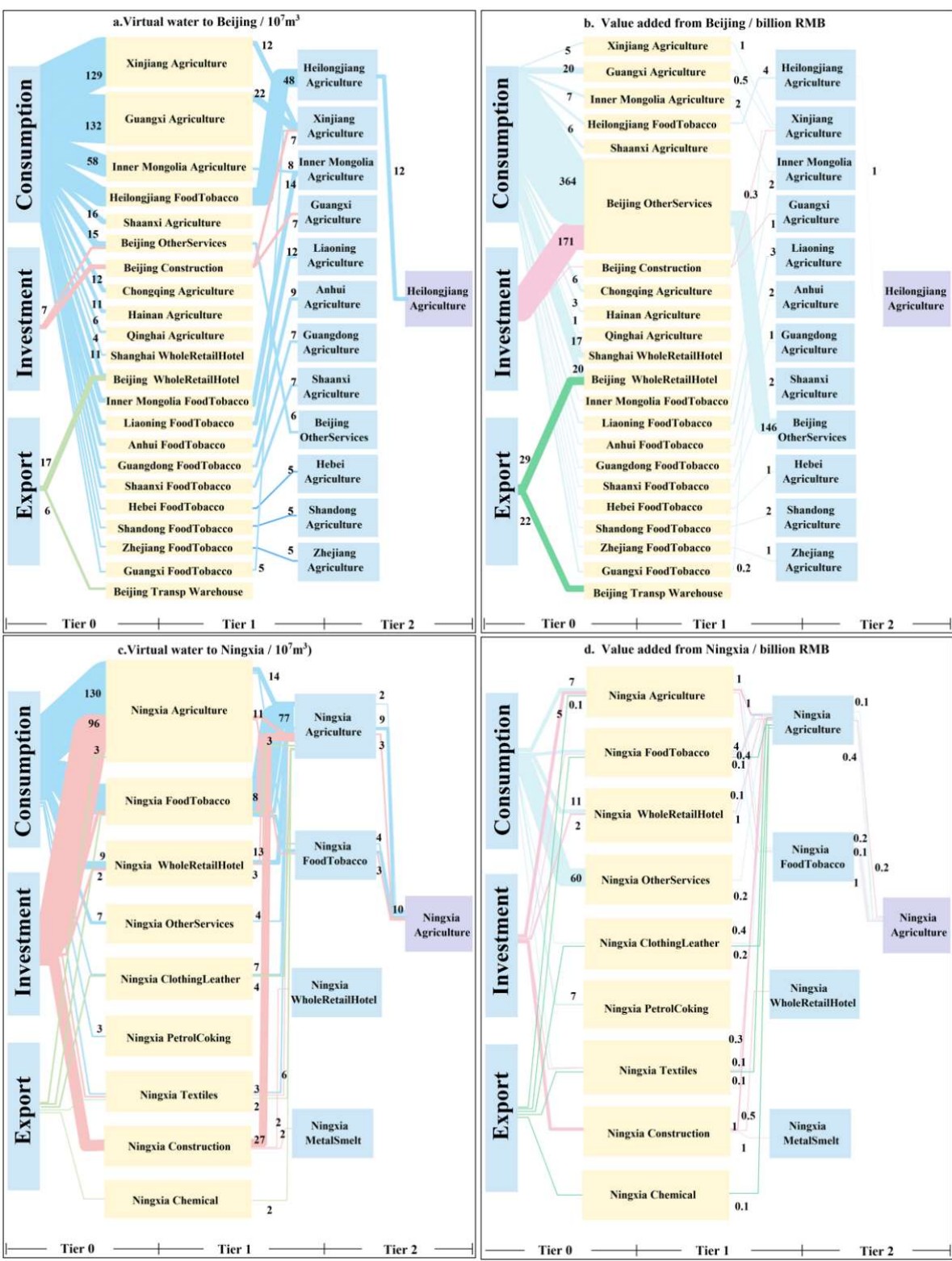

**Figure 8.** (**a**,**b**) show the 30 top ranking virtual water supply chain paths and value added for Beijing in 2017. (**c**,**d**) show the key virtual water supply chain paths and value added for Ningxia in 2017.

The main virtual water supply paths of Beijing were from agriculture in Guangxi, Xinjiang, and Inner Mongolia directly to the consumption of Beijing and from agriculture in Heilongjiang to food and tobacco in Heilongjiang, and then to Beijing consumption demand. Conversely, the economic benefits transferred on these main paths were far less than those transferred directly from services in Beijing or through production tier transfer to consumption and investment demand. The four top ranking virtual water supply paths of Beijing, Guangxi Agriculture → Beijing Consumption, Xinjiang Agriculture → Beijing Consumption, Inner Mongolia Agriculture → Beijing Consumption, and Heilongjiang Agriculture → Heilongjiang Food and Tobacco → Beijing Consumption, accounted for 28.22% of the total virtual water supplied to Beijing, but the value added transferred on the paths only accounted for 1.44%. Among the 30 top ranking virtual water supply paths, the three top ranking value-added paths were Beijing Services → Beijing Consumption, Beijing Services → Beijing Investment, and Beijing Services → Beijing Services → Beijing Consumption, accounting for only 2.49% of the virtual water supplied to Beijing, but the value added transferred on the paths accounted for as much as 33.21%.

The main virtual water supply paths of Ningxia were from agriculture in Ningxia directly to consumption and investment demand, and from agriculture in Ningxia to food and tobacco in Ningxia to consumption demand. Moreover, Ningxia acquired only a small portion of value added on the main virtual water supply paths. The four top ranking virtual water supply paths of Ningxia were Ningxia Agriculture → Ningxia Consumption, Ningxia Agriculture → Ningxia Investment, Ningxia Agriculture → Ningxia Food and Tobacco → Ningxia Consumption, and Ningxia Agriculture → Ningxia Construction → Ningxia Investment, accounting for 48.6% of the total virtual water supplied to Ningxia, but the value added transferred on the paths only accounted for 2.93%. Among the 30 top ranking virtual water supply paths, the three top ranking value-added paths were Ningxia Service → Ningxia Consumption, Ningxia Wholesale and Retail Catering → Ningxia Consumption, and Ningxia Agriculture → Ningxia Consumption, accounting for 21.49% of the total virtual water supplied to Ningxia, but the value added transferred on the paths only accounted for 13.29%.

## 4. Conclusions and Policy Implications

Given the prominent differences in water resource endowments and economic conditions among Chinese provinces, customized water resource protection policies and economic development strategies should be formulated for different provinces. This study analyzed the net transfers of virtual water and value added across provinces in China, which could be used as a reference to develop strategic, differentiated water conservation and water-pricing policies. Due to the close and complex links among China's provinces, this study used MRIO and SPA models to track virtual water and value added along supply chains. The results and policy implications are summarized below.

First, the developed provinces located in the south coast, east coast, and Beijing–Tianjin have alleviated trade-related water shortages, whereas the developing provinces located in the northwest, southwest, and northeast have exacerbated water shortages due to trade. For example, in 2017, Guangdong, Zhejiang, and Beijing had 16, 12.4, and 9.4 Gm$^3$ net inflows of virtual water, respectively, in interprovincial trade, accounting for about 54% of the total inflows of virtual water between provinces in China. However, Xinjiang, Guangxi, and Heilongjiang had 26.9, 13.3, and 7.5 Gm$^3$ net outflows of virtual water in interprovincial trade, accounting for 68% of the total outflows of virtual water between provinces in China. These developing provinces must formulate water conservation policies to improve water-use efficiency in production processes, such as differentiated water prices to encourage enterprises to adopt more advanced water-saving technologies [54]. In addition, water abstraction licenses and water rights could also protect water-deficient provinces [55]. Such water-saving policies should also comprehensively consider the virtual water supply chain paths of China's provinces [20]. Our results demonstrated that the main supply chain paths of virtual water transferred from upstream to downstream sectors in intermediate

production tiers were short. The T0 tier was identified as the main tier of interprovincial virtual water transfers in China, followed by the T1 tier. Therefore, reducing one-step or two-step water use in upstream sectors can decrease virtual water consumption in downstream sectors. For example, Guangxi, Xinjiang, and Inner Mongolia Agriculture → Beijing Consumption and Heilongjiang Agriculture → Heilongjiang Food and Tobacco → Beijing Consumption were the four top ranking virtual water supply paths of Beijing, and Ningxia Agriculture → Ningxia Consumption and Investment, Ningxia Agriculture → Ningxia Food and Tobacco → Ningxia Consumption, and Ningxia Agriculture → Ningxia Construction → Ningxia Investment were the four top ranking virtual water supply paths of Ningxia. Improving the water-use efficiency of these key virtual water supply chain paths can cultivate a high-efficiency virtual water supply chain.

Second, the developed provinces located in the north, on the east coast, and in Beijing–Tianjin have gained economic benefits due to trade, whereas the developing provinces located in the northwest, central, and southwest regions have suffered economic losses due to trade. For example, in 2017, Shandong, Jiangsu, and Beijing had 885, 494, and 302 billion RMB net inflows of value added in interprovincial trade, accounting for 43% of the total inflows of interprovincial value added; however, Xinjiang, Henan, and Yunnan had 462, 896, and 729 billion RMB net outflows of value added in interprovincial trade, accounting for 53% of the total outflows of interprovincial value added. Therefore, the developing provinces located in the northwest, central, and southwest regions should cultivate effective domestic and foreign demand, accelerate industrial trade-related transformation, and coordinate development with the developed provinces in the north, on the east coast, and in Beijing–Tianjin. The developing provinces located in the central and northwest regions have a considerable proportion of the nation's natural resources and labor force. If these resources are effectively and rationally leveraged to form advantageous industries and develop characteristic economies, the dilemma of low labor costs and serious virtual water consumption may be broken. Furthermore, if the industries in the developed provinces located in the north, on the east coast, and in Beijing–Tianjin are transferred to the developing provinces, it would help to accelerate economic structure adjustments, reduce the time of industrial upgrade, and promote coordinated development among the provinces, narrowing the existing gaps between the developed provinces and developing provinces demonstrated in this study.

Third, the developed provinces were the primary beneficiaries of the net transfers of virtual water and value added through China's interprovincial trade, whereas the developing provinces in the northwest and central regions suffered from considerable imbalances in virtual water and economic benefits transfers. For example, Beijing in bilateral trade with Xinjiang (WVI = 2.06) and Hubei (WVI = 1.62) not only had 2.48 and 0.05 $Gm^3$ net inflows of virtual water, but also 14.6 and 35.7 billion RMB net inflows of value added, respectively. In this regard, virtual water compensation schemes following a principle of "who benefits, who compensates" may adjust such imbalances [56]. The beneficiary province can compensate the province suffering from inequality according to imbalances in the degree of net virtual water and value-added transfer (WVI) and net virtual water flow in bilateral trade. For example, Beijing could compensate Xinjiang and Hubei by the product of the net virtual water flow volume and a compensation price based on WVI value. Having considered the virtual water supply chain paths, the virtual water flowing to the developed provinces on the key paths was primarily supplied by other provinces, whereas the developing provinces were supplied internally, and the economic benefits transferred on these key paths were imbalanced. For example, all four of the top ranking virtual water supply chain paths of Beijing came from other provinces, accounting for 28.22% of the total virtual water flowing to Beijing, but the value added only accounted for 1.44%, whereas all four of the top ranking virtual water supply chain paths of Ningxia were supplied internally, accounting for 48.6% of the total virtual water flowing to Ningxia, but the value added only accounted for 2.93%. Therefore, it is suggested to provide subsidies

to the nodes on these key supply paths to reduce the imbalance between virtual water and economic benefit transfers.

**Author Contributions:** Conceptualization, J.Z. and S.W.; methodology, J.Z.; software, J.Z.; validation, J.Z., S.W. and Q.L.; formal analysis, S.W.; investigation, Q.L.; resources, S.L.; data curation, Q.A.; writing—original draft preparation, J.Z.; writing—review and editing, L.L.; visualization, J.Z.; supervision, S.W.; project administration, S.W.; funding acquisition, S.W. All authors have read and agreed to the published version of the manuscript.

**Funding:** This research was supported by the National Natural Science Foundation of China under Grant Nos. 71773118, and 71733003.

**Institutional Review Board Statement:** Not applicable.

**Informed Consent Statement:** Not applicable.

**Data Availability Statement:** The datasets generated and/or analyzed during the current study are available from the corresponding author upon reasonable request.

**Conflicts of Interest:** The authors declare no conflict of interest.

## Appendix A

**Table A1.** Classification of China's 30 Provinces.

| No. | Name of Provinces or Cities | Region |
|---|---|---|
| 1 | Beijing | Beijing–Tianjin |
| 2 | Tianjin | |
| 3 | Hebei | north |
| 4 | Shandong | |
| 5 | Liaoning | northeast |
| 6 | Jilin | |
| 7 | Heilongjiang | |
| 8 | Shanghai | east coast |
| 9 | Jiangsu | |
| 10 | Zhejiang | |
| 11 | Fujian | south coast |
| 12 | Guangdong | |
| 13 | Hainan | |
| 14 | Anhui | central |
| 15 | Shanxi | |
| 16 | Jiangxi | |
| 17 | Henan | |
| 18 | Hubei | |
| 19 | Hunan | |
| 20 | Guangxi | southwest |
| 21 | Chongqing | |
| 22 | Sichuan | |
| 23 | Guizhou | |
| 24 | Yunnan | |
| 25 | Inner Mongolia | northwest |
| 26 | Shaanxi | |
| 27 | Gansu | |
| 28 | Qinghai | |
| 29 | Ningxia | |
| 30 | Xinjiang | |

## Appendix B

**Table A2.** The Name Abbreviations of the 28 Sectors Considered in This Study.

| No. | Name of Sectors | Name Abb. |
|---|---|---|
| 1 | Agriculture, forestry, and fishing | Agriculture |
| 2 | Coal mining | CoalMining |
| 3 | Mining of petroleum and natural gas | CrudeOilGas |
| 4 | Metal ores mining | MetalOre |
| 5 | Non-metallic minerals mining | NonmetalOre |
| 6 | Production of food and tobacco | FoodTobacco |
| 7 | Textiles | Textiles |
| 8 | Wearing apparel, leather, fur, etc. | ClothingLeather |
| 9 | Wood processing and furniture | WoodFurniture |
| 10 | Papermaking, printing, stationery, etc. | PaperPrinting |
| 11 | Fossil fuel refining | PetrolCoking |
| 12 | Chemical industry | Chemical |
| 13 | Production of non-metallic mineral products | NonmetalProducts |
| 14 | Smelting and processing of metals | MetalSmelt |
| 15 | Metal products | MetalProducts |
| 16 | General and special equipment | GeneralSpecialEquip |
| 17 | Transport equipment | TransportEquip |
| 18 | Electrical equipment | ElectricalEquip |
| 19 | Electronic equipment | ElectronicEquip |
| 20 | Measuring instruments and meters | MeasureInstru |
| 21 | Other processing products | OtherProc |
| 22 | Electricity and heat power | ElectricHeatpower |
| 23 | Gas supply | GasSupply |
| 24 | Water supply | WaterSupply |
| 25 | Construction | Construction |
| 26 | Transport and warehousing | TranspWarehouse |
| 27 | Wholesale, retail, hotels, and catering | WholeRetailHotel |
| 28 | Other services | OtherServices |

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
