# Peer review of "Revealing Virtual Water Transfers and Imbalanced Economic Benefits Hidden in China’s Interprovincial Trade"

_water, doi:10.3390/w14111677_

Round 1

Reviewer 1 Report

Abstract:

Lines 30-32:  "It is suggested that provinces adopt differentiated water-use systems and provide subsidies to the nodes on key virtual water supply chain paths." State WHY, and to what WHAT HOPED FOR EFFECT.

Introduction:

To alleviate shortages and imbalanced distribution of water resources in China, the government has initiated more than 20 water conservancy projects, such as the South-to-North Water Transfer Project  IS THIS REALLY A WATER CONSERVANCY PROJECT - OR RATHER A PHYSICAL WATER TRANSFER PROJECT?, and has implemented three stringent controlling red lines concerning water use (Liu et al., 2013).

Methodology:

clear and transparent

Results and Discussion:

clear and detailed

Conclusion and Policy Implications:

clear and appropriate

References:

[1] Allan, J. A., 1997. ‘Virtual water’: a long-term solution for water short Middle Eastern economies? School of Oriental and Asian Studies, University of London, pp. 24–29.

[9] Helpman, E., Krugman, P., 1987. Market structure and foreign trade: Increasing returns, imperfect competition, and the international economy. MIT Press: Cambridge, MA.

[13] Leontief, W., 1986. Input-output economics. 2nd ed. Oxford University Press: Oxford.

Reviewer 2 Report

The paper is very informative. Nevertheless, it looks more like a technical report, rather than a sci paper. The authors should focus on the novelties and not the application of a well-known methodology.

The abstract is too detailed. Cut the long story short...
Keep only what is absolutely important and attractive for the reader.

avoid first plural throughout the entire manuscript. Apply where necessary.

Additional comments:

1. What is the main question addressed by the research?  Virtual water interprovincial trading in China. 

2. Do you consider the topic original or relevant in the field? Does it address a specific gap in the field? NO

3. What does it add to the subject area compared with other published material? Nothing new in my opinion

4. What specific improvements should the authors consider regarding the methodology? What further controls should be considered?
Nothing

5. Are the conclusions consistent with the evidence and arguments presented and do they address the main question posed?
YES
6. Are the references appropriate? YES

7. Please include any additional comments on the tables and figures. Several Figures even though they are quite small (e.g.2; 5; 6; 7), they are not inappropriate. Nevertheless the editor should decide whether they should be enlarged or not
